# Temporal Enhancement for Video Affective Content Analysis

## ABSTRACT

With the popularity and advancement of the Internet and video-sharing platforms, video affective content analysis has been greatly developed. Nevertheless, existing methods often utilize simple models to extract semantic information. This might not capture comprehensive emotional cues in videos. In addition, these methods tend to overlook the presence of substantial irrelevant information in videos, as well as the uneven importance of modalities for emotional tasks. This could result in noise from both temporal fragments and modalities, thus diminishing the capability of the model to identify crucial temporal fragments and recognize emotions. To tackle the above issues, in this paper, we propose a Temporal Enhancement (TE) method. Specifically, we employ three encoders for extracting features at various levels and sample features to enhance temporal data, thereby enriching video representation and improving the model's robustness to noise. Subsequently, we design a cross-modal temporal enhancement module to enhance temporal information for every modal feature. This module interacts with multiple modalities at once to emphasize critical temporal fragments while suppressing irrelevant ones. The experimental results on four benchmark datasets show that the proposed temporal enhancement method achieves state-of-the-art performance in video affective content analysis. Moreover, the effectiveness of each module is confirmed through ablation experiments.

## CCS CONCEPTS

• **Computing methodologies → Artificial intelligence**.

## KEYWORDS

video affective content analysis, temporal enhancement, cross-modal attention

## 1 INTRODUCTION

In today's digital era, video serves as a primary medium for conveying information. It enriches individuals' daily lives and social interactions with its unique narrative style and visual expression. With the widespread adoption of the Internet and the popularity of video-sharing platforms, a vast array of video content emerges, emphasizing the crucial necessity for accurately comprehending and analyzing the emotional undertones within videos. Video affective content analysis seeks to capture various levels of semantic information, including the video's core content and depicted events, to understand and predict the emotions that the video is expected

Unpublished working draft. Not for distribution.

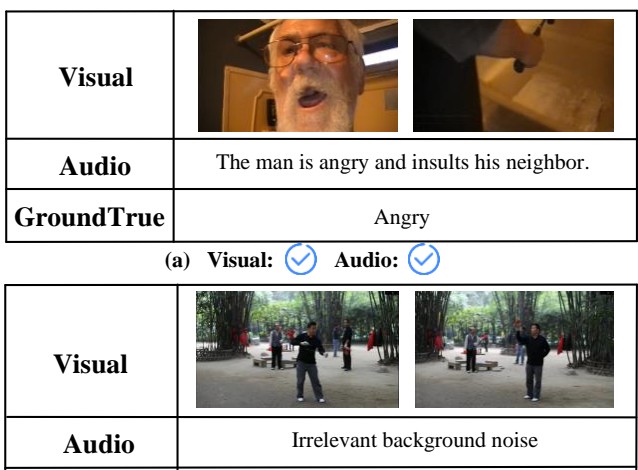

|  | |
|---|---|
| **Visual** | |
| **Audio** | The man is angry and insults his neighbor. |
| **GroundTrue** | Angry |

(a) **Visual:** ✓ **Audio:** ✓

|  | |
|---|---|
| **Visual** | |
| **Audio** | Irrelevant background noise |
| **GroundTrue** | Joy |

(b) **Visual:** ✓ **Audio:** ✗

**Figure 1: Illustrations on whether modalities in different videos can provide clues for emotional labels.**

to induce in viewers[23, 36]. It holds significance across diverse domains, including personalized video recommendations, emotional robotics, and the evaluation of film and television productions.

There are currently two challenges in the task of video affective content analysis. One concerns accurately characterizing the information content within video and audio modalities, while the other involves efficiently integrating the distinctive task-related information provided by different modalities and capturing critical temporal fragments for emotional cues. Regarding the first challenge, deep learning-based feature extraction[16, 32, 34, 35] is increasingly employed. However, these approaches often utilize simplistic models for extracting action or scene information from visual data, which may not sufficiently capture the complete semantic information of the video. Building upon this, Pan et al.[15] introduced the CLIP[17] to encode static single-frame image semantic information, aiming to attain enhanced semantic representation. Yet, It has not fully leveraged the potent performance of the CLIP encoder. Addressing the second challenge, two methods are currently prevalent in this field including late fusion[35] and feature fusion[15, 34]. The former treats different modalities as independent, with each passing through distinct data pipelines, thereby lacking interconnectivity between them. In contrast, the latter facilitates modality interaction during the model's feature representation learning process, leading to superior practical performance. Of course, as the number of modalities (information sources) increases, designing effective fusion modules introduces new challenges.

However, in the current research, the presence of noise in various types of information is often overlooked, leading to suboptimal outcomes in models. Noise may arise from two sources. First, in

video affective content analysis datasets, video durations are typically long, while the core clips capable of inducing groundtruth emotions have relatively short durations[16]. This undoubtedly results in more noise in terms of the temporal dimension that is unrelated to the groundtruth emotion. Second, certain modalities also function as noise compared to others. As illustrated in Fig1, the visual information depicts "The man couldn't stand the noise of the neighbor's decoration, so he goes to the neighbor's house to insult and smashes the bathtub". At the same time, the audio captures "The man is angry and insults the neighbor" information in Fig 1(a). Both visual and audio modalities offer precise and effective emotional cues. Thus, the model should simultaneously consider information from both modalities during the learning process. Nevertheless, in Fig1(b), the visual description portrays "The man is performing acrobatics," while the audio provides only "Irrelevant background noise." During learning, the model should prioritize visual temporal information and suppress audio temporal information. Current methods treat each modality as equally important when different modalities interact, thereby introducing noise at the modality level.

To address the aforementioned issues about noise, we propose a video affective content analysis method based on temporal enhancement. First, to represent complete semantic information in the video, we employ three encoders to extract different feature attributes such as semantics, motions, and audio characteristics. Subsequently, we perform a feature temporal sampling to enrich the temporal representation of the video, enhancing temporal data and ensuring the model's robustness to noise. Second, acknowledging the varying importance of different modalities during modal interaction, we design a cross-modal temporal enhancement module that accepts inputs of various modal features and designates one modality as the primary modality for interaction with others. Its aim is to focus on task-relevant temporal fragments while mitigating the impact of irrelevant ones. In this process, the temporal features of modalities mutually influence each other rather than operating independently. Finally, to enable synchronous interaction among all modal temporal fragments, a global Transformer with shared weights integrates cross-modal and intra-modal interaction.

We conduct experiments on the widely used VideoEmotion-8[26], YF-E6[9], and LIRIS-ACCEDE[2] datasets, along with the newly introduced VAD[24] dataset. The experimental results indicate that our method outperforms state-of-the-art methods and showcases its superiority. Additionally, the ablation experiments confirm the significance of each module in our method.

In summary, the contributions of this paper are as follows:

- We design a cross-modal temporal enhancement module. It facilitates the transfer of information from other modalities to the primary modality, thereby augmenting the temporal sequence of the primary modality. This module takes into consideration the significance of each temporal fragment while amalgamating temporal data from all modalities.
- We present a simple methodological framework named Temporal Enhancement (TE). It can fortify both temporal representation and temporal information, ultimately enhancing the model's resilience to noise and the ability to detect and recognize key temporal fragments.

- The effectiveness of the proposed method is validated through a lot of experiments and further analyses conducted on four distinct datasets.

## 2 RELATED WORKS

### 2.1 Video Affective Content Analysis

Video affective content analysis aims to predict the emotions expected to be elicited from viewers by the video. The key lies in capturing the core visual or acoustic temporal information and mapping it to the emotional space. Early research mainly focused on designing representative and useful features for identifying highly abstract emotions. Jiang et al.[9] employed numerous low-level and intermediate features, such as SBank and OBank[10], for identifying emotions. Sikka et al.[19] simply combined multiple visual audio features to conduct emotions classification on video fragments. Qiu et al.[16] pieced together action and scene features as a whole, and then inputted them into a dual attention network to learn frame information related to emotions and reduce the influence of irrelevant information.

Recently, compared to manually selected features, deep learning-based features and methods have demonstrated excellent representation and predictive ability in predicting emotions in videos [15, 16, 30, 32, 34, 35]. Zhang et al.[32] extracted frame-level depth features and used discrete Fourier transform to obtain kernel features for emotion recognition. Qiu et al.[16] proposed a dual-focus attention network (DFAN), in which the time series focus module concentrated on temporal keyframes, while the frame object focus module concentrated on objects in each frame and searches for the key objects that best represent emotional labels. Zhao et al.[35] were the first to propose an end-to-end deep network named VAANet, which applies spatial, channel, and temporal attention to visual features and temporal attention to audio features for recognizing video emotions. Zhang et al.[34] believed that previous methods mainly focused on key visual frames, which may limit the ability to encode the context describing expected emotions. Therefore, a temporal erasure network was proposed, which locates keyframes in a weakly supervised manner and can also learn contextual information. Pan et al.[15] used three different video encoders to extract features, namely visual image and motion features, as well as emotional features of audio. They designed a cross-temporal multimodal fusion module for temporal interaction within and between modalities to capture temporal relationships between different modalities. Simultaneously, to address the issue of insufficient model supervision by a single emotional label, the use of time-synchronized comments as auxiliary supervision are proposed to provide richer emotional clues.

Numerous works solely extract action and scene data from visual modalities, potentially insufficient for accurately representing video content, thereby fundamentally constraining emotion prediction abilities. Additionally, they frequently disregard the presence of emotion-independent noise in the video, further compromising the model's capacity to capture crucial temporal fragments. Even though Pan et al. [15] utilized different encoders for visual modalities, it did not fully use the powerful semantic understanding of the CLIP model due to the use of continuous fragments. Therefore, we propose to use multiple encoders while utilizing temporal feature

sampling to enrich the semantic representation of the video and make the model more robust to noise.

## 2.2 Multimodal Fusion

The purpose of multimodal fusion is to integrate information from two or more modalities to improve the prediction accuracy[1]. Traditional multimodal fusion uses either early fusion or late fusion methods, which involve simple concatenation or addition[6]. Gradually, more fusion strategies are being implemented from the perspective of feature interaction. Some studies [11, 27, 31] treat each modal feature as a tensor, transforming modal fusion into tensor fusion. Zadeh et al.[31] proposed the tensor fusion layer (TFN) to simulate single-modal, bimodal, and bimodal interactions. Liu et al. [11] proposed the low-rank multimodal fusion method (LMF) to address the issue of poor computational efficiency in tensor fusion. Multimodal sequences often exhibit discontinuous properties, requiring the inference of long-term dependencies across modalities. Due to the success of Transformer models in these areas, some studies are based on Transformer models to handle multimodal fusion[3, 12, 20, 21]. Tsai et al.[21] proposed MulT, which extends the standard Transformer framework into a multimodal Transformer model to directly fuse misaligned multimodal data. Cheng et al.[12] argued that independent pairwise fusion in MulT cannot utilize the advanced features of the source modality. Thus, they proposed a progressive modal reinforcement method (PMR) for multimodal fusion of misaligned multimodal sequences.

In tasks involving video affective content analysis, the fusion of information encompasses various modal features from both video and audio sources. Given the inherent temporal nature of videos, modal interaction methods based on the transformer or attention mechanisms are extensively employed in this field. Zhang et al. [34] introduced a temporal correlation learning module to comprehensively explore implicit correspondences among different fragments across audio and visual modalities. Pan et al.[15] devised a cross-temporal multimodal fusion module that utilized self-attention to understand the interrelations between modalities within each video fragment and across different fragments. This enables full capture of temporal dependencies between visual and audio signals. However, Each modality can only independently focus on useful clues from other modalities, without considering the differences in importance of these modalities. Because temporal fragments that can be given greater weight in a certain modality may have a very limited effect when placed in the perspective of all modalities. Treating all modalities equally may introduce noise, indirectly weakening the role of the most crucial temporal cues. Therefore, we propose a cross-modal temporal enhancement module. When each modality is enhanced, the information from multiple modalities is combined to enhance the correlated temporal fragments, while weakening the weight of irrelevant temporal fragments to suppress noise.

## 3 PROBELM DESCRIPTION

The purpose of video affective content analysis is to utilize the semantic information present in both video and audio to predict the emotional responses evoked in viewers. This entails the model learning a mapping from the semantic space to the emotional space. The input consists of the original video denoted as $x_v \in R^{T_v \times H \times W \times 3}$

and the original audio represented as $x_a \in R^{l_a}$. Here, $T_v$, $H$, and $W$ symbolize the video frame number, as well as the height and width of an individual frame image, respectively. Additionally, $l_a$ signifies the data length of a one-dimensional audio signal. In classification tasks, the output $\hat{y}$ represents a discrete category, signifying a particular emotion. Conversely, in regression tasks, $\hat{y}$ denotes a continuous real value, indicating the intensity and magnitude of a specific emotional dimension.

## 4 METHODOLOGY

The overall framework of video affective content analysis based on temporal enhancement method is illustrated in Fig 2. The two modalities, video, and audio, undergo feature extraction, yielding various semantic features via three encoders. Initially, preliminary enhancement of temporal data is conducted through feature temporal sampling. Subsequently, the three features undergo independent cross-modal temporal enhancement via separate modules, resulting in enhanced temporal information for each modal feature. Following this, the three features are concatenated along the temporal dimension, and fed into the global Transformer to facilitate temporal synchronization interaction across all modalities. Lastly, the fully integrated features undergo temporal averaging to represent the entire feature, and then they are concatenated and inputted into the classifier for emotion prediction.

### 4.1 Feature Extraction

**Fragment Motion Features:** To extract fragment motion features from the original pixel video, the video must first be segmented. Here, each fragment comprises 16 frames, with an overlap of 8 frames between adjacent fragments. Thus, for a video with the frame number of $T_v$, the total number of fragments is $T_m = T_v/8$. Resizing each frame of the fragment to $112 \times 112$ directly, the 3D ResNet[7] is employed to extract motion features independently from each fragment, and outputs feature $f_m^i \in \mathbb{R}^{H_m \times W_m \times C_m}$. As we do not focus on spatial information details, the output features are spatially averaged to obtain vector representations of the fragments. Finally, the motion features extracted from each video are represented as $f_m \in \mathbb{R}^{T_m \times C_m}$.

**Image Semantic Features:** To maintain alignment with motion features, a single frame of the video is selected every 8 frames, and each frame is scaled to a size of $224 \times 224$. For each frame, we utilize the basic CLIP model (ViT-B/32)[17] as the image encoder. The vector at the [CLS] token serves as the feature representation of the entire image, with a dimension of $f_s^i \in \mathbb{R}^{C_s}$. Finally, the semantic features extracted from each video are represented as $f_s \in \mathbb{R}^{T_s \times C_s}$, where $T_s$ is calculated as $T_v/8$.

**Audio Characteristic Features**: Initially, Mel-Frequency Cepstral Coefficients (MFCC) are employed to derive the audio characteristic description from the original audio signal. By establishing a proportional relationship, the temporal interval of each motion fragment is mapped to the corresponding temporal position of MFCC features, thereby acquiring the primary features of the associated audio fragment. These primary features are fed into the audio recognition model VGGish[1], which has been pre-trained on the AudioSet dataset[5], to extract the high-level feature representation of the

---

[1]https://github.com/JMGaljaard/VGGish-pytorch

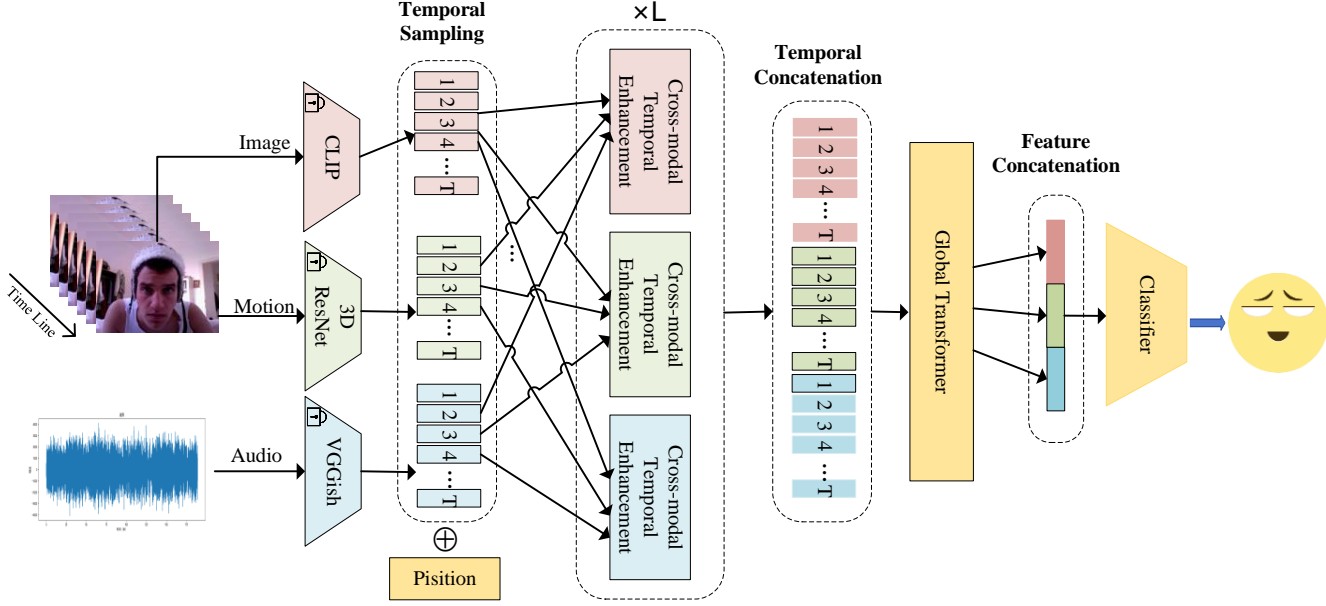

**Figure 2: Illustration of the overall structure of the temporal enhancement method. The various modal features are distinguished by different colors. Locked modules do not participate in training. As data progresses through the pipeline, the temporal information from each modality is enhanced.**

audio fragment. The model outputs the feature vector $f_a^i \in \mathbb{R}^{C_a}$, and ultimately, the temporal feature representation of the entire audio signal is denoted as $f_a \in \mathbb{R}^{T_a \times C_a}$.

The encoders responsible for extracting various semantic features are not involved in the training process. Consequently, in practical applications, these extracted features are pre-stored, leading to a significant reduction in computational workload and acceleration of the training process.

### 4.2 Temporal Sampling

During the feature extraction, each feature maintains equal temporal length and adheres to a strict one-to-one correspondence, denoted as $T_m = T_s = T_a = T_v/8 = T_c$, where $T_c$ represents the common length. However, $T_c$ tends to vary significantly among different videos within the dataset, posing challenges for model training. Furthermore, videos often contain multiple fragments that evoke emotional responses in viewers. They are distributed across various periods. Simultaneously, numerous fragments may lack relevance to the groudtruth label. Consequently, the model needs to effectively identify key temporal fragments while disregarding noise fragments. The strategy of segmenting continuous fragments in [15] generally requires setting a long temporal length to encompass useful fragments. However, this may result in a surplus of redundant and homogeneous data during training, hindering the model's ability to discern temporal feature differences and potentially leading to overfitting.

Unlike the strategy in [15], we opt to randomly sample the entire temporal features directly, without requiring continuity. The main motivation for this choice is that the CLIP model is powerful

enough for semantic understanding and representation capabilities of single-frame images. In addition, the representation of videos from different perspectives can be greatly enriched. These representations are temporally distinct, which provides an opportunity for the model to notice key temporal fragments. Specifically, the three encoded and stored features have a common temporal length of $T_c$. We define a constant sampling temporal length, denoted as $T$. We utilize a random sampler from the set $\{1, 2, 3, \cdots, T_c\}$, sampling $T$ times without replacement to obtain $T$ unique numbers, as depicted in following equation:

$$[id_1, id_2, \cdot, id_T] = Sampler(\{1, 2, 3, \cdots, T_c\}) \quad (1)$$

After that, the $T$ numbers are sorted from small to large to simulate their temporal sequence:

$$[s_1, s_2, \cdots, s_T] = Sorted([id_1, id_2, \cdots, id_T]) \quad (2)$$

Indexing among the three features according to the sorted numbers to obtain the representation of all features of the video and audio modalities in the current batch:

$$\begin{aligned}
\bar{f}_m &= f_m[s_1, s_2, \cdots, s_T] \in \mathbb{R}^{T \times C_m} \\
\bar{f}_s &= f_s[s_1, s_2, \cdots, s_T] \in \mathbb{R}^{T \times C_s} \\
\bar{f}_a &= f_a[s_1, s_2, \cdots, s_T] \in \mathbb{R}^{T \times C_a}
\end{aligned} \quad (3)$$

For the subsequent interaction of the three features, three linear layers are used to map the features to the same dimension $d$. The transformed features are $\bar{f}_m, \bar{f}_s, \bar{f}_a \in R^{T \times d}$. Since temporal sampling does not consider the temporal relationship between fragments, we choose absolute position encoding to inject sequential information into the temporal features. The position encoding utilizes sine and

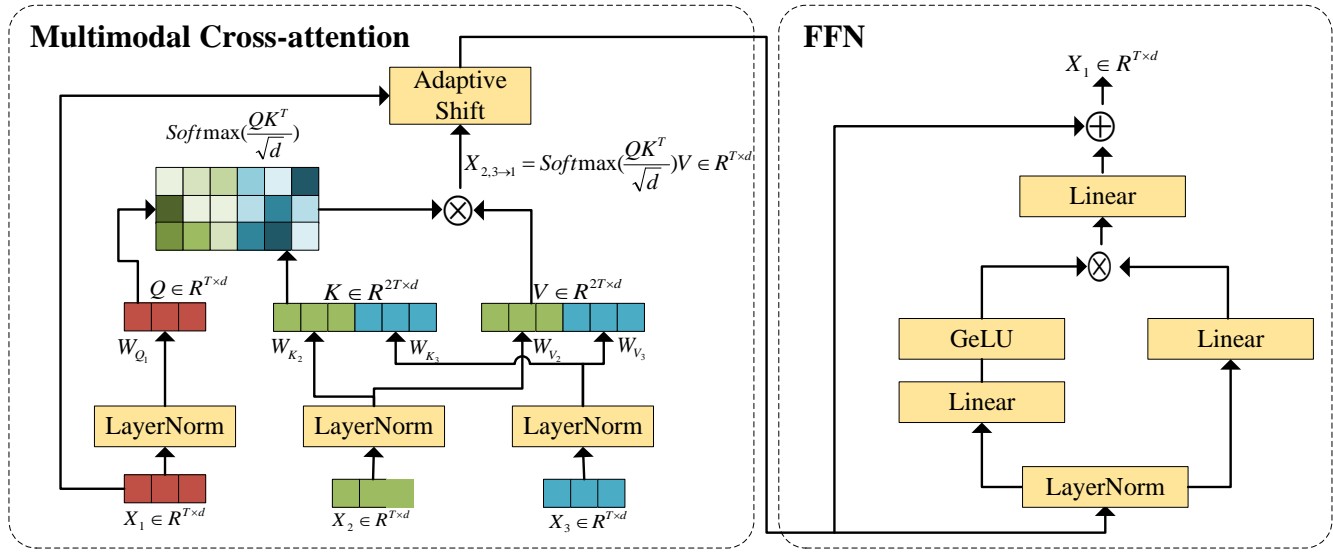

**Figure 3: The cross-modal temporal enhancement module consists of multimodal cross attention and feed-forward network (FFN).**

cosine functions of different frequencies. Its formula is as follows:

$$PE_{(x,2y)} = sin(\frac{x}{10000^{2y/d}})$$
$$PE_{(x,2y+1)} = cos(\frac{x}{10000^{2y/d}}) \tag{4}$$

where $x$, $y$ and $d$ respectively represent the temporal position, the position of the dimension, and the size of the dimension. The calculated fixed position coding value is added to the three transformed features to obtain the final temporal feature representation.

### 4.3 Cross-Modal Temporal Enhancement Module

In order to enhance the temporal information of each feature, we design a cross-modal temporal enhancement module based on the attention mechanism. The module enables one modality to incorporate information from other modalities effectively. As illustrated in Fig 3, this module comprises two main components. First, multimodal cross-attention facilitates the simultaneous integration of various modal features, focusing on relevant temporal fragments while suppressing irrelevant ones. Additionally, to standardize the numerical range of diverse features, the adaptive shift module is employed at the residual connection. Second, a variant of the gated linear unit (GLU) introduced by Dauphin et al.[18] is utilized to construct the FFN, thereby facilitating the nonlinear transformation of modal features.

*4.3.1 Multimodal Cross-Attention.* Assume that there are three modal temporal feature inputs $X_1, X_2, X_3 \in \mathbb{R}^{T \times d}$, where $X_1$ is the feature that needs to be enhanced, and the others are features that provide information. First, layer regularization is performed on each feature to obtain a normalized representation:

$$X_1^{ln} = LN_1(X_1); X_2^{ln} = LN_2(X_2); X_3^{ln} = LN_3(X_3) \tag{5}$$

then $X_1^{ln}$ is mapped to the query domain, and $X_2^{ln}, X_3^{ln}$ is mapped to the key domain and the value domain through linear transformation:

$$Q = X_1^{ln} W_{Q_1}$$
$$K_2 = X_2^{ln} W_{K_2}; V_2 = X_2^{ln} W_{V_2} \tag{6}$$
$$K_3 = X_3^{ln} W_{K_3}; V_3 = X_3^{ln} W_{V_3}$$

where $W_{Q_1}, W_{K_2}, W_{V_2}, W_{K_3}, W_{V_3} \in R^{d \times d}$ is the learnable parameter matrix. $K_2$ and $K_3$ are spliced along temporal dimension to form the multimodal key $K \in \mathbb{R}^{2T \times d}$, and $V_2$ and $V_3$ are spliced along temporal dimension to form a multimodal value $V \in \mathbb{R}^{2T \times d}$. The attention weight between the multimodal key $K$ and the query $Q$ is multiplied with the multimodal value $V$ to obtain the enhanced feature representation, as followings:

$$X_{2,3 \to 1} = softmax(\frac{QK^T}{\sqrt{d}})V \in R^{T \times d} \tag{7}$$

When querying keys of different modalities, it's important to note that the same query domain is utilized. During the calculation of attention weights for all query-key pairs, each temporal fragment is compared across all modalities. This process accounts for the intrinsic importance of each modality. Even less significant fragments within a modality are given relatively higher weights, these fragments may still compete with important fragments from other modalities at a global level, resulting in them ultimately receiving small weights.

In order to integrate the features $X_{2,3 \to 1}$ obtained by information transfer into the original features $X_1$, we use the scaling factor $\alpha$ to limit the numerical adaptation of the temporal feature of the former to a reasonable range, the calculation formula is as follows:

$$\bar{X}_1 = X_1 + \alpha X_{2,3 \to 1} \tag{8}$$

$$\alpha = min\{\frac{\|X_1\|_2}{\|X_{2,3\rightarrow1}\|_2}, 1\} \quad (9)$$

where $\|\cdot\|_2$ represents the L2 norm.

*4.3.2 FFN.* Unlike the general feed-forward network, this module has two branches, with additional branches serving as gating mechanisms. The primary function of this module is to apply a specific nonlinear transformation to the enhanced features $\bar{X}_1$. The features undergo layer regularization before being separately fed into the two branches. The outputs of these branches are element-wise multiplied. Then the result is subjected to linear transformation and added to the original input, resulting in the final enhanced feature representation. The formula is as follows:

$$\bar{X}_1^{ln} = LN(\bar{X}_1)$$
$$X_1 = \bar{X}_1 + (GeLU(X_1^{\bar{l}n}W_1) \otimes (X_1^{\bar{l}n}W_2))W_3 \quad (10)$$

where $W_1, W_2, W_3$ represents the matrix parameters in the linear layer, and $GeLU$ is the activation function.

## 4.4 Global Transformer and Prediction

The three modal features undergo multiple layers of independent cross-modal temporal enhancement modules, continuously reinforcing each feature until the modalities achieve full interaction. However, the temporal fragments within the modalities have yet to establish a connection with each other. To address this, a global Transformer with shared matrix parameters is employed to model all temporal relationships within and between modalities simultaneously. The enhanced features obtained above are denoted as $\bar{f}_m, \bar{f}_s, \bar{f}_a$ respectively. They are then concatenated along the temporal dimension to obtain a multimodal feature representation $\bar{f} = [\bar{f}_m; \bar{f}_s; \bar{f}_a]$. Initially, all temporal interactions are completed through the self-attention layer, as demonstrated in the following:

$$\bar{f}^{ln} = LN(\bar{f})$$
$$\bar{f} = \bar{f}^{ln} + Softmax(\frac{(\bar{f}^{ln}W_Q)(\bar{f}^{ln}W_K)^T}{\sqrt{d}})(\bar{f}^{ln}W_V) \quad (11)$$

where $W_Q, W_K, Q_V$ represents the three learnable parameter matrices, which act on three modalities at the same time. Subsequently, the temporal features are transformed nonlinearly through the general feedforward network layer, as following:

$$\bar{f}^{ln} = LN(\bar{f})$$
$$\bar{f} = \bar{f}^{ln} + ReLU(\bar{f}^{ln}W_1)W_2 \quad (12)$$

where $W_1, W_2$ represents the two learnable parameters in the feed-forward network.

After the interaction, the multimodal features are split into their respective modal features $[\bar{f}_m; \bar{f}_s; \bar{f}_a] = \bar{f}$ in the temporal dimension. These features are averaged to represent the final vector representation of semantic features at different levels. Finally, the feature vectors are spliced and inputted to the classifier/regressor to complete emotion prediction. The calculation formula is as follows:

$$\hat{y} = MLP([\bar{f}_m^{mean}, \bar{f}_s^{mean}, \bar{f}_a^{mean}]) \quad (13)$$

where MLP stands for the classifier/regressor, which is a nonlinear multi-layer perceptron.

## 5 EXPERIMENTS

This section first introduces the dataset used in the experiment, and then explains the technical details. After that, we compare our method with state-of-the-art approaches. Finally, we conduct the ablation experiment to verify the effectiveness of the features and modules. It is worth mentioning that we further analyze the training process of the temporal enhancement method and the role of cross-modal temporal enhancement through visualization in the supplementary material.

### 5.1 Datasets

To evaluate the proposed method based on temporal enhancement, we employ commonly used public datasets in the field of video affective content analysis, including VideoEmotion-8[9], YF-E6[26], LIRIS-ACCEDE [2]. In addition, the new VAD dataset[24] has the same number of modal features as we extracted in the above three datasets, so our method is also experimented on the VAD dataset and compared with its baseline model. A detailed introduction to each dataset as well as the experimental setup is included in the supplementary material.

### 5.2 Implementation Details

Based on different label attributes, the datasets are categorized into two groups, each requiring specific evaluation metrics. For the VideoEmotion-8, YF-E6, and VAD datasets, they involve classification tasks. So the Accuracy (ACC) serves as the primary evaluation metric and the F1-score is supplemented for the VAD dataset. In contrast, for the MediaEval2016 task[13] utilizing the LIRIS-ACCEDE dataset, it entails a regression task. So the Mean Squared Error (MSE) and Pearson Correlation Coefficient (PCC) are employed as evaluation metrics. Other relevant implementation details are included in the supplementary material.

### 5.3 Comparison with state-of-the-art methods

We compare the proposed method with state-of-the-art methods on various datasets in the field of video affective content analysis. For YF-E6 and VideoEmotion-8 datasets, advanced works include VAANet[35], Dual[16], TAM[15], WECL[34], KeyFrame[25], and FAEIL[33]. They are all based on deep learning methods. For the MediaEval2016 task, the state-of-the-art methods include RBN[4], MMDRBN[22], MML[28], AFRN[29], MMLGAN[14] and TAM[15]. They include both traditional and deep learning based models. For the VAD dataset, we compare with the baseline models provided in the dataset, including TFN[31], MISA[8], and MulT[21].

Table 1 compares our method with the advanced approaches on the YF-E6 and VideoEmotion-8 datasets. In the table, all methods are further differentiated according to the two attributes of "additional auxiliary data" and "pre-stored features". As illustrated in Table 1, our method achieves the classification accuracy of 64.91% and 59.39% on the YF-E6 and VideoEmotion-8 datasets respectively, which are improved by 3.91% and 1.76% compared to the previous best results. This demonstrates the effectiveness of the proposed temporal enhancement method. To exclude the improvement brought about solely by adding the image semantic information of the CLIP encoder, our method is compared with the "TAM w/o TSC" method. Both use the same encoder and share the pre-stored

features without the participation of additional data. Our method shows an improvement of 4.27% and 3.35% on the two datasets respectively. Even though the TAM method adds additional auxiliary data, our method still far exceeds it. Compared with the KeyFrame and FAEIL methods that use additional auxiliary data, our method surpasses them by extracting information about emotional clues and events in the original video through a powerful encoder. This approach proves to be more effective than the way that uses additional data but lets the model learn from the original signal. It also demonstrates that there is still untapped potential in the original video. It is worth mentioning that although the method of pre-stored features cannot perform spatial data enhancement on single-frame images in the video like VAANet and WECL methods, and may lead to a slight loss in performance, it significantly accelerates model training and prediction.

Table 2 compares our method with the state-of-the-art approaches on the MediaEval2016 task. In the Valence regression task, our method achieves the best performance on both indicators. Specifically, compared with the previous best method, our method reduced the MSE metric by 0.004 while increasing the PCC metric by 0.028. In the Arousal regression task, our method achieves only moderate results on the MSE, but again attains the highest results on the PCC. Pan et al.[15] explain the reasons for this phenomenon. That is, the numerical distribution of the Arousal label is highly concentrated. Even if the model predicts a constant value of 3.3, it can still yield the MSE of 0.6159, which is also much higher than the result of our method. However, the model obviously did not learn any emotional information about the input, so the PCC metric may be more suitable to evaluate this task[13]. It is worth noting that, when comparing our method with the "TAM w/o TSC" method, our method outperforms it in both metrics, with improvements of 0.2 and 0.038 on the MSE and PCC respectively. This improvement shows that even for regression tasks, our method based on temporal enhancement is more effective than the cross-modal temporal fusion method proposed in TAM. Combining the results of Table 1 and Table 2, our method outperforms the advanced approaches in both classification and regression tasks. This indicates the excellent ability of our method to capture and identify task-related core temporal information.

Table 3 shows the experimental results of our method and baseline models on the new VAD dataset. As observed from the table, our method outperforms other models on five of the six metrics for the three labels, with the remaining metric deviating from the best results by only 0.2%. Concerning Valence and Arousal labels, the MulT stands out as the best-performing model. Similar to our method, both employ cross-modal attention. However, unlike our method, the modal interactions in MulT are independent of each other and solely receive information from the original signal. Contrarily, our method receives information from all modalities simultaneously during each modal interaction. This enables the concentration of the most relevant information between modalities while suppressing all weakly relevant information through the softmax function. In comparison to MulT, our method improves the accuracy metric by 1.5% and 1.2% respectively. For the primary emotion labels, our method compared with MISA achieves improvements of 3.7% and 4.2% in accuracy and F1-score respectively. In comparison to MulT, although our method does not attain the highest value in the

**Table 1: Comparison with the state-of-the-art methods on YF-E6 and VideoEmotion-8 datasets. The result represents the accuracy (%) on the test set.**

| Method | Auxiliary | Pre-store | YF-E6 | VideoEmotion-8 |
|---|---|---|---|---|
| VAANet | ✗ | ✗ | 54.5 | 55.3 |
| Dual | ✗ | ✓ | 57.37 | 53.34 |
| TAM w/o TSC | ✗ | ✓ | 60.64 | 56.04 |
| WECL | ✗ | ✗ | 58.2 | 57.3 |
| KeyFrame | ✓ | ✗ | 59.51 | 52.85 |
| FAEIL | ✓ | ✗ | 60.44 | 57.63 |
| TAM | ✓ | ✓ | 61.00 | 57.53 |
| **TE** | ✗ | ✓ | **64.91** | **59.39** |

**Table 2: Comparison with the state-of-the-art methods on the MediaEval2016 task.**

| | MediaEval2016 | | | |
|---|---|---|---|---|
| | Valence | | Arousal | |
| Method | MSE↓ | PCC↑ | MSE↓ | PCC↑ |
| RBN | 0.332 | 0.387 | 0.766 | 0.416 |
| MMDRBN | 0.303 | 0.405 | 0.713 | 0.470 |
| MML | 0.198 | 0.399 | 1.173 | 0.446 |
| MMLGAN | 0.194 | 0.445 | 1.077 | 0.491 |
| AFRN | 0.193 | 0.468 | **0.524** | 0.522 |
| TAM w/o TSC | 0.172 | 0.529 | 1.115 | 0.550 |
| TAM | 0.177 | 0.533 | 0.754 | 0.560 |
| **TE** | **0.168** | **0.561** | 0.915 | **0.588** |

**Table 3: Comparison with the baseline models on the new VAD dataset.**

| | VAD | | | | | |
|---|---|---|---|---|---|---|
| | Valence | | Arousal | | Primary Emotion | |
| Method | ACC↑ | F1↑ | ACC↑ | F1↑ | ACC↑ | F1↑ |
| TFN | 0.630 | 0.626 | 0.603 | 0.585 | 0.273 | 0.206 |
| MISA | 0.633 | 0.631 | 0.593 | 0.564 | 0.449 | 0.216 |
| MulT | 0.644 | 0.642 | 0.618 | 0.608 | 0.338 | **0.260** |
| **TE** | **0.659** | **0.657** | **0.630** | **0.612** | **0.486** | 0.258 |

F1-score, the F1-score metric only decreases by 0.2%, while the accuracy surges by 14.8%. This phenomenon is explained in [24]. MISA prioritizes predicting categories with a large proportion, whereas MulT focuses on predicting categories with a smaller proportion. Undoubtedly, our method achieves a better balance between the two and yields commendable results.

## 5.4 Ablation Study

In order to explore the importance of different features for video affective content analysis and the effect of different feature combinations, we conduct ablation experiments on motion features, image semantic features, and audio features on the YF-E6 dataset. The experimental results are shown in Table4. Comparing the results of all combinations of only two features, the "image + audio" combination achieves the best results, while the "motion + audio" combination

**Table 4: Ablation experiment of each feature on YF-E6 dataset. The result represents the accuracy (%) on the test set.**

| Motion | Image | Audio | YF-E6 |
|:---:|:---:|:---:|:---:|
| ✓ | ✓ | ✗ | 61.12 |
| ✗ | ✓ | ✓ | 64.43 |
| ✓ | ✗ | ✓ | 59.05 |
| ✓ | ✓ | ✓ | **64.91** |

**Table 5: Ablation experiments of each module on YF-E6 dataset. The result represents the accuracy (%) on the test set.**

| Method | YF-E6 |
|:---|:---:|
| Simple Concatenate | 61.49 |
| Overall Method | **64.91** |
| w/o position encoding | 64.79 |
| w/o global Transformer | 64.30 |
| w/o cross-modal temporal enhancement | 63.20 |
| independent modal interaction | 64.18 |

achieves the worst results. Although both image features and motion features are visual features and are extracted from videos, the results show that image features can provide more emotional clues than motion features, which may benefit from the powerful semantic understanding ability of the CLIP encoder. The combination of only two features can be regarded as the discarding of a certain feature from the complete feature combination. Comparing each two-feature combination with the complete feature combination and observing their performance degradation, the importance of the feature can be roughly estimated. Their importance from high to low is image, audio, and motion. Although the performance of the best two-feature combination "image + audio" is very close to that of the complete feature combination, the results still show slight improvement after introducing the relatively less important motion feature. This reflects the nature of multimodal learning, whereby by integrating features with different characteristics, the model can learn the complementary aspects of different features to make accurate predictions.

In order to verify the effectiveness of each module in our method, we conduct ablation experiments by removing components from the overall method. The results of all experiments are presented in Table5. The "Simple Concatenation" method refers to taking the temporal average of each feature and splicing them for classification after feature sampling, bypassing the intermediate modality interaction process. The "independent modal interaction" method involves replacing the method of combining multiple modal features simultaneously in the cross-modal temporal enhancement module with the way of independent modal interaction in [15, 21]. The former focuses on the salient parts of all modal features and suppresses the relatively ineffective parts, while the latter focuses on important fragments in each modality separately, treating all modalities interacting with a certain modality as equally important. First, regarding the "Simple Concatenation" method, it achieved an accuracy of 61.49%. Comparing this to the best 61.00% obtained by the previous method on the YF-E6 dataset in Table1, it is evident that

temporal data enhancement achieved by simple feature sampling yields promising results during training. Building upon this, coupled with subsequent cross-modal temporal enhancement modules and other modules, the "Overall Method" further improves accuracy by 3.42% compared to "Simple Concatenation". These demonstrate the effectiveness of the temporal enhancement method. Second, "w/o" denotes removing a module from the overall method. Removing the position coding has little impact on emotion prediction, resulting in only a 0.12% decrease in performance. Since features are sampled along the temporal dimension, whether to add position coding affects the results minimally. Removing the global Transformer results in a 0.61% decrease in performance, emphasizing the importance of simultaneous inter-modal and intra-modal interaction. Removing the cross-modal temporal enhancement module leads to a sharp drop in accuracy by 1.71%. Even though a global Transformer for intra-modal and inter-modal interaction still exists, the transformation matrix of each feature is shared at this point, limiting the ability of each modality to learn unique information and utilize the complementary advantages of different modalities fully. This illustrates the importance of the cross-modal temporal enhancement module. Finally, replacing the interaction method in the cross-modal temporal enhancement module with independent modal interaction decreases recognition accuracy by 0.73%. This may be because the model assigns certain temporal fragments relatively unimportant large weight without integrating all modal temporal information, introducing unnecessary noise and weakening performance.

## 5.5 Conclusion

In this paper, we propose a method for video affective content analysis based on temporal enhancement in order to solve the impact of noise from both temporal and modal aspects. The method utilizes three encoders to extract various modal features and conducts temporal data enhancement via temporal feature sampling. Then, we design the cross-modal temporal enhancement module to facilitate temporal interaction among modalities, emphasizing significant temporal fragments across all modalities during the interaction. Finally, the global Transformer is employed to facilitate both intra-modal and inter-modal interaction across all modes. Comparisons between our method and existing advanced approaches on various datasets demonstrate the superiority of our method. The significance of each modal feature and module in our method is revealed through ablation experiments.

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
