# OpenReview forum: "Temporal Enhancement for Video Affective Content Analysis"
_acmmm.org/ACMMM/2024/Conference — MM2024 Oral_

### Official Review · Reviewer_F9U7 · 2024-05-22

**Rating:** 4
**Confidence:** 3

**Summary:**

This paper explores the importance of different features for video affective content analysis. The proposed method enhances temporal representation and information resilience to noise. The core module is a gated cross-modal attention to process the importance of each modality. Experiments are conducted following [15].

**Strengths:**

1.	The paper is well-structured.
2.	Extensive experiments demonstrate the effectiveness of proposed method in both classification and regression tasks.
3.	Ablation studies verify that each component is an optimal solution.

**Limitations:**

1.	The idea of leveraging relevant information within a video is widely explored in the field. For instance, [15] employs Danmu-based mask modeling to capture important affective parts, while [A] identifies the most critical part and incorporates them into a dual-branch network for accurate recognition. Additionally, [B] proposes a method to localize temporal affective segments of the video. It is crucial for the authors to discuss the differences between their proposed method and these similar works to provide clarity. This comparison will help reviewers better understand the unique contributions and advancements of this work, distinguishing it from existing approaches.
2.	The proposed method has not achieved consistent performance in arousal regression on the MedEval dataset and emotion classification on the VAD dataset. And the paper does not provide an explanation for these inconsistencies. It is essential to analyze the differences across datasets and provide both qualitative and quantitative results to better understand the underlying reasons for the varied performance. Such an analysis would offer valuable insights into the strengths and limitations of the proposed method, helping to identify areas for improvement and ensuring a more comprehensive evaluation of its effectiveness.

[A] Contextual Causal Intervention Module, CVPR23

[B] Temporal Sentiment Localization: Listen and Look in Untrimmed Videos, ACM MM22

**Suitability:**

3

---

### Official Review · Reviewer_h5zk · 2024-05-26

**Rating:** 4
**Confidence:** 2

**Summary:**

This paper presents a method for video affective content analysis that is based on temporal enhancement. The semantic information of the video is modeled using three encoders that are associated with three different characteristics; visual semantics, motions and audio. Then, a temporal sampling is performed to enrich the temporal representation of the video, to ensure the method's robustness to noise. The varying importance of the different data modalities for each different video sample is taken into account, and handled using a cross-modal temporal enhancement module that gets all the different data modalities and indicates which one is the primary and how it will interact with the remaining ones. Synchronous interaction among the data modalities is enabled using a global Transformer with shared weights that integrates cross- and intra-modal interaction. The performance of this method is evaluated using several benchmarking datasets and metrics, and compared with various SoA approaches. Ablations indicate the contribution of the different data modalities and network components, in the overall performance.

**Strengths:**

The paper is well-written and easy to ready. The task is described clearly, and the same goes for the literature review. The proposed methodology is presented in a well-structured manner, with the help of descriptive illustrations. The performance of the method is extensively evaluated using various benchmarking datasets and evaluation protocols from the literature, and compared with many SoA approaches. The reported comparisons point out the advanced performance of the proposed method, while the authors discuss also the reasons for the weak performance of this method in one of the experimental settings. The conducted ablations quantify the contribution of each data modality and network/processing component, in the overall performance. Extra implementation details and experimental results are provided and (qualitatively) discussed in the supplementary material.

**Limitations:**

My main concern relates to the novelty of this paper. The proposed method exhibits a significant level of similarity with the work in [15]. The use of a different encoder (CLIP) for modeling data does not constitute a novelty imho. Any major difference resides in the temporal sampling process and the weighted contribution of the different data modalities in the data fusion process; though both approaches can be seen as variants/extensions of the corresponding processes in [15]. I am not sure if this level of novelty is sufficient for publication in MM.

Other than that, the paper needs proof reading to fix typos and grammar errors; e.g. "Yet, It has", "Even though Pan et al. [15] utilized..., it did not fully use", "Probelm description", "However, Each modality", "Pisition" in Figure 2

**Suitability:**

3

---

### Official Review · Reviewer_qDc8 · 2024-06-01

**Rating:** 5
**Confidence:** 4

**Summary:**

This paper proposes a multimodal method for video affective content analysis, addressing issues related to modality-specific noisy data by designing temporal representation enhancement and selection that increases resilience to noise. The paper targets a thorough representation of the video, analyzing features that represent semantics, motion and audio characteristics. Overall the paper is well written and the approach is well described and well tested.

**Strengths:**

1. The methods are thoroughly tested and analyzed, experiments being done on four different datasets and ablation studies being performed in order to better describe performance and individual module contribution.

2. The motivation for this particular approach is clearly stated and analyzed.

3. I particularly appreciate the thorough way multimodality is addressed in this paper, and the attention the design of the method pays to the way modalities interact.

**Limitations:**

1. Some English presentation problems and typos in the paper. Please make sure to give the paper close inspection with regards to this in case of publication. The ones I could find are:
- Figure 1: should be Ground Truth instead of Ground True;
- Page 3, row 245: “single-modal, bimodal, and bimodal interactions” - bimodal twice;
- Page 3, row 271: “However, Each modality” – capital letter;
- Page 3, row 273: sentence starting with “Because temporal …” seems to be incomplete or lack a predicate;
- Page 3, row 321: “for a video with the frame number of T_v” – I think it is rather total frame count T_v;

2. Section 3 – Problem Description. The paper mentions, at the end of this section, a comparison between discrete versus continuous representation of emotions / affective content. It is my understanding that this refers to representations of concrete emotions (data with labels like disgust, happiness, etc.) versus representations of emotions in spaces like valence-arousal. I believe this should be stated more clearly.

3. Missing references:
- Page 3, row 345: besides the link to VGGish implementation, a reference to the paper must also be given to: CNN Architectures for Large-Scale Audio Classification, Shawn Hershey et al.

**Suitability:**

3

---

### Official Review · Reviewer_8y2t · 2024-06-01

**Rating:** 5
**Confidence:** 4

**Summary:**

This paper describes a method for joint audio-visual emotion recognition in videos. The key innovation is the cross-modal temporal enhancement which learns the relationship between the various modalities so as to learn how to suppress irrelevant modalities when necessary. The proposed method equals the state of the art and through ablation shows the efficacy of the various modules. The paper is well written and the experimentation is thorough.

**Strengths:**

1. Well written paper with some typos such as "Probelm" in the section heading for section 3.
2. Interesting new approach to irrelevant modality rejection.
3. Decent experimental results. Thorough experimentation. Interesting ablation.
4. Thorough literature survey.

**Limitations:**

1. Other than the proposed cross-modal enhancement, the novelty of the paper is light. Not a serious weakness in my view.
2. The ablation seems to show only a modest improvement because of the cross-modal enhancement. So while the method is interesting and sound, it is not that it offers a substantial improvement over the state of the art. The authors need to explain this well.

**Suitability:**

3

---

### Official Review · Reviewer_LKJF · 2024-06-08

**Rating:** 4
**Confidence:** 3

**Summary:**

This paper proposes a Temporal Enhancement (TE) method that facilitates the transfer of information from other modalities to the primary modality, thereby augmenting the temporal sequence of the primary modality. Extensive experiments and further analyses were conducted on four distinct datasets.

**Strengths:**

The method appears very logical, and the experiments were conducted using four public datasets. The results seem quite convincing.

**Limitations:**

1. The experiments in Tables 4 and 5 were conducted using only one dataset. Are the conclusions consistent with those from VideoEmotion-8?
2. The implementation details of the Global Transformer are not clearly explained. Providing a more detailed description would be beneficial.
3. It would be useful to explain whether the audio input was processed using a sliding window to generate clips, which were then passed through VGGish for feature extraction before performing Temporal Sampling.

**Suitability:**

3

---

### Meta-Review · Area_Chair_yZas · 2024-07-03

**Recommendation:** Accept (Oral)
**Confidence:** 5

**Metareview:**

The reviewers were all positive about this work. The authors also addressed several concerns of the reviewers. Given the number of reviews and the positive impact, I would recommend this for an oral presentation.